# Mutual Interplay of Host Immune System and Gut Microbiota in the Immunopathology of Atherosclerosis

**DOI:** 10.3390/ijms21228729

**Published:** 2020-11-19

**Authors:** Chih-Fan Yeh, Ying-Hsien Chen, Sheng-Fu Liu, Hsien-Li Kao, Ming-Shiang Wu, Kai-Chien Yang, Wei-Kai Wu

**Affiliations:** 1Division of Cardiology, Department of Internal Medicine and Cardiovascular Center, National Taiwan University Hospital, Taipei 100, Taiwan; nicholas.yeh@gmail.com (C.-F.Y.); indiglo314@gmail.com (Y.-H.C.); hsienli_kao@yahoo.com (H.-L.K.); 2Department and Graduate Institute of Pharmacology, College of Medicine, National Taiwan University, Taipei 100, Taiwan; 3Department of Internal Medicine, National Taiwan University Hospital, Hsin-Chu Biomedical Park Hospital, Hsin-Chu 30261, Taiwan; liewliu0818@gmail.com; 4Department of Internal Medicine, National Taiwan University Hospital, Taipei 100, Taiwan; mingshiang@ntu.edu.tw; 5Research Center for Developmental Biology & Regenerative Medicine, National Taiwan University, Taipei 10617, Taiwan; 6Institute of Biomedical Sciences, Academia Sinica, Taipei 11529, Taiwan; 7Department of Medical Research, National Taiwan University Hospital, Taipei 100, Taiwan

**Keywords:** gut microbiota, atherosclerosis, dysbiosis, immune system, metabolites

## Abstract

Inflammation is the key for the initiation and progression of atherosclerosis. Accumulating evidence has revealed that an altered gut microbiome (dysbiosis) triggers both local and systemic inflammation to cause chronic inflammatory diseases, including atherosclerosis. There have been some microbiome-relevant pro-inflammatory mechanisms proposed to link the relationships between dysbiosis and atherosclerosis such as gut permeability disruption, trigger of innate immunity from lipopolysaccharide (LPS), and generation of proatherogenic metabolites, such as trimethylamine N-oxide (TMAO). Meanwhile, immune responses, such as inflammasome activation and cytokine production, could reshape both composition and function of the microbiota. In fact, the immune system delicately modulates the interplay between microbiota and atherogenesis. Recent clinical trials have suggested the potential of immunomodulation as a treatment strategy of atherosclerosis. Here in this review, we present current knowledge regarding to the roles of microbiota in contributing atherosclerotic pathogenesis and highlight translational perspectives by discussing the mutual interplay between microbiota and immune system on atherogenesis.

## 1. Introduction

Atherosclerosis is an arterial disease featured with subendothelial accumulation of plaque in the arterial wall and narrowing of the lumen. Atherosclerosis can result in numerous cardiovascular complications, including stroke and coronary artery disease (CAD), all of which are associated with significant morbidity and mortality. Atherogenesis is a complex, multistep process that remains incompletely understood [1]. Macrophage activation and foam cell formation have been demonstrated to play important roles in the plaque formation in different stages of vascular inflammation [2]. A growing body of evidence suggests that dysbiosis, a state of microbiota imbalance in the gut, contributes critically to the development of vascular inflammation [3,4]. In this regard, the gut microbiota, now considered as a dynamic endocrine organ, interacts with the host at intestinal surface and sends messages to distant organs through numerous microbially generated mediators, including multiple microbial products and several bacteria-induced cytokines [5].

The atherogenic inflammatory state can be initiated by an unhealthy gut microbiota and promoted by subsequent dysregulation of local and systemic immune responses. The macrophages residing in intestinal lamina propria and intima layer of artery can be activated through the reception of several types of signals from gut microbiota, including translocated live bacteria, structural components of dead bacteria (e.g., lipopolysaccharides (LPS), peptidoglycan, flagellin, and bacterial DNA), and microbially-derived functional metabolites (e.g., short-chain fatty acids (SCFAs), bile acids (BAs), and trimethylamine N-oxide (TMAO)) [6]. These microbially-derived small molecules can be transmitted from the gut lumen into the body through transcellular or paracellular routes while larger molecules such as LPS may be translocated through a disrupted gut barrier. The immune cells react to these bacterial signals locally and systemically and produce various cytokines and protein—such as IL-1, IL-17, IL-18, IL-22, antimicrobial peptides, and immunoglobulin A (IgA)—to direct a defensive response or an immune tolerance to the gut microbiota [7]. Furthermore, immune cells primed in the intestine by the gut microbiota may migrate to distal vessels and organs to induce immunopathogenesis [8]. Therefore, the harmony between the host and its gut bacteria serves to balance the pro- and anti-inflammatory states [9]. Given that the gut microbiota is an important and manipulable environmental mediator of human health, understanding the atherosclerotic immunopathology induced by imbalanced host–microbe interactions may lead to the identification of novel targets for the prevention and treatment of atherosclerotic cardiovascular diseases [10]. In this review, we discuss the potential immunopathology of the atherosclerosis contributed by the consequences of mutual interplay between the gut microbiota and gut immune system as well as the atherogenesis in plaque triggered by microbially-generated products.

## 2. Roles of the Oral and Gut Microbiota in Cardiovascular Diseases (CVD)

As the largest endocrine organ, the gut microbiota participates in and facilitates numerous physiological processes. Its associations with metabolic conditions such as obesity, type 2 diabetes, and atherosclerosis have been well-established [11]. Early studies demonstrating associations of the gut microbiota with obesity and insulin resistance laid the foundation for further investigations on metabolic disorders and atherosclerosis [12]. The contribution of the gut microbiota to atherosclerosis was further supported by the presence of bacterial DNA in human atherosclerotic plaque, where bacterial DNA abundance correlated with their amounts in the oral cavity [13]. Other studies have reported differences in gut microbial communities between patients with symptomatic atherosclerosis and healthy controls [14,15,16,17]. In addition, the abundance of certain microorganisms has been found to predict CVD risk [14,15,16,17]. This evidence has opened new avenues for research on gut–heart and gut–vessel connections.

Accumulating evidence has shown the gut microbiota affects atherogenic susceptibility. It was noted that atherosclerotic lesion formation could be significantly attenuated in germ-free (GF), compared to conventionally raised, *ApoE^−/−^* mice [18]. This was supported by reduced thrombogenicity of conventionally raised *Tlr2^−/−^* mice compared to wild-type mice, while no difference in thrombus formation in GF model after carotid artery injury [19]. On the other hand, a balanced commensal gut bacterial community could play a protective role on atherosclerosis development [20]. It was reported that the conventionally raised mice has significantly reduced atherosclerotic plaques than germ-free mice while feeding with normal chow diet [21]. Therefore, the imbalanced status of gut microbiota, so called dysbiosis, has been considered as a risk factor for atherosclerosis development and progression; and could be independent to traditional risk factors such as cigarette smoking, dyslipidemia, diabetes mellitus, and hypertension. Brown et al. constructed a model delineating the role of the gut flora in the progression of atherosclerotic CVD [22]. In addition, several mechanisms linking dysbiosis and atherosclerosis have been postulated (Figure 1). These include leaky gut syndrome, which is associated with inflammation, and cardiometabolic phenotypes associated with the translocation of microbiota-mediated metabolites or dysregulation of bile acid metabolism. Specifically, leaky gut syndrome refers to the impairment and increased permeability of the intestinal mucosal barrier, which allows the entry of LPS and peptidoglycan, membrane components of Gram-negative bacteria, into the bloodstream. Immune cells recognize LPS by utilizing Toll-like receptors (TLRs) to induce inflammation through the activation of pro-inflammatory cytokine production [23]. The presence of these bacterial components in the bloodstream can cause hyperlipidemia, insulin resistance, and vascular inflammation, all of which can elicit atherogenesis [22].

Microbiota mainly affects atherosclerosis by two major pathways. First, immune system activated by bacterial translocation and structural components of bacteria induces inflammatory response. Bacteria from gut or oral cavity invade the atherosclerotic plaque directly or induce proatherogenic response by translocation of microbe-associated molecular pattern (MAMPs), such as lipopolysaccharide (LPS). This would increase the production of proinflammatory cytokines and chemokines and further aggravate atherosclerosis. Second, specific microbial metabolism of dietary components can lead to the production of both beneficial and harmful small molecules. These metabolites may contribute to the development of atherosclerotic plaques or augment the disease. For examples, trimethylamine N-oxide (TMAO), a microbiota-derived metabolite from choline and carnitine, inhibit reverse cholesterol transport (RCT), and induce systemic inflammation. Both phenylacetylglutamine (PAGln), a microbiota-derived metabolite from phenylalanine, and TMAO induce platelet hyperactivity to increase thrombus formation. On the other hand, the short-chain fatty acids (SCFAs), derived from dietary fiber fermentation, enhance gut barrier and reduce systemic inflammation, both of which inhibit atherogenesis.

The gut microbiota is also a critical component of the host metabolism, producing or facilitating the production of numerous essential metabolites, of which some (e.g., TMAO, SCFAs, and phenylacetylglutamine (PAGln)) have been related to the pathogenesis of atherosclerosis. TMAO, a metabolite produced through microbial metabolism of dietary carnitine and choline, has been well established for its contribution to the progression of atherosclerosis by either indirectly supplementing choline/carnitine or directly adding TMAO to the diet of mice models [24,25]. TMAO has been shown to increase atherosclerotic plaque formation and progression and promote platelet hyperresponsiveness, thrombosis, and vascular inflammation. One study reported a strong association between high fasting plasma TMAO level and major adverse cardiovascular events (MACE) [26]. The gut microbiota also regulates the metabolism of dietary fibers and essential amino acids by breaking down dietary fiber to produce SCFAs (e.g., acetate, propionate, and butyrate) that control blood pressure and participate in cardiac repair following myocardial infarction [27,28]. PAGln, a metaorganismal metabolite derived from phenylalanine metabolism, was recently found to facilitate platelet hyperresponsiveness and thrombosis [29]. In addition, studies have indicated associations of high plasma PAGln with increased risks of CVD and MACE [29,30]. Besides, protein-bound uremic toxins p-Cresyl sulfate (pCS) and indoxyl sulfate, metabolized from tyrosine and tryptophan, respectively [31,32], have been reported to cause renal fibrosis by increasing production of reactive oxygen species in renal tubular epithelial cells and were demonstrated to be associated with CVD and mortality in patients with chronic kidney disease [33,34,35]. In addition to producing metabolites from dietary components, the gut microbiota converts primary BAs to secondary BAs through 7α dihydroxylation and bile salt hydrolysis. BAs regulate numerous metabolic pathways, such as lipid and glucose metabolism through interaction with nuclear receptor farnesoid X (FXR) and membrane receptor Takeda G-protein–coupled receptor 5 (TGR5) which involve the pathways to inhibit or promote inflammation and atherogenesis [36,37,38]. These studies have supported the premise that the gut microbiota critically regulates atherogenesis.

Associations between the oral microbiota and atherosclerosis development have also been described. A meta-analysis of seven cohorts indicated higher CVD risk in people with periodontal diseases such as periodontitis, tooth loss, gingivitis, and bone loss [39]. Other studies have established that poor dental hygiene increases the risks of CVD and acute myocardial infarction [40,41,42]. Bacterial DNA of *Aggregatibacter actinomycetemcomitans* and *Tannerella forsythia* (previously *Actinobacillus actinomycetemcomitans* and *Tannerella forsythensis*) as well as *Porphyromonas gingivalis* have been detected in atheromatous plaque, suggesting that these oral pathogens could reach vascular subendothelium and increase the accumulation of atherosclerotic plaque [43,44]. These findings also imply that periodontal pathogens present in atherosclerotic plaque may lead to the development of atherosclerosis and CVD.

The strong association between dysbiosis and atherosclerosis indicates the potential of gut microbiota manipulation as a treatment option for atherosclerosis. Current strategies include dietary intervention, fecal microbiota transplantation (FMT), and the administration of antimicrobials, bacterial enzyme inhibitors, prebiotics or probiotics, and host enzyme inhibitors which have yet to demonstrate therapeutic efficacy for atherosclerosis in human study. Antimicrobial treatment is regarded as a potential method for intervening the gut microbiota. In both mice and humans, oral antibiotics suppress the gut microbiota, leading to significant reductions in plasma trimethylamine (TMA) and TMAO. However, they have only been reported to ameliorate TMA- or TMAO-induced atherosclerotic phenotypes in mice [25,26,45]. In addition to the inconclusive effectiveness in human studies and the risk of serious side effects, antimicrobial treatment for atherosclerosis may not be feasible for the potential emergence of multidrug-resistant strains. These findings indicate the complexity of gut–host interactions and a better understanding of the immunopathogenesis between microbiota and host provides insight into the development of new therapeutic targets.

## 3. Role of Inflammation in Atherosclerosis

Atherosclerosis, as an inflammatory disease, is characterized by monocyte and macrophage infiltration into the arterial wall and the expression of pro-inflammatory cytokines [46,47]. The impacts of both innate and adaptive immune responses on atherosclerosis have been well-documented [48]. Monocytes, macrophages, and cytokines are critical participants in such reactions.

Macrophages exhibit high functional plasticity and induce or inhibit inflammation in response to various cytokines and microbial products [49]. In-depth fate-mapping and lineage-tracing studies have been reported to discover four types of macrophage populations in tissues during steady or diseased states, including tissue-resident, self-renewing macrophages; monocyte-derived inflammatory macrophages; exclusively monocyte-derived constitutive tissue macrophages (non-self-renewing); and monocytes migrating through tissues [50]. Both tissue-resident macrophages and monocyte-derived macrophages are vital in initiating and maintaining inflammatory responses. Atherosclerosis develops through a multiphasic numerical escalation of the monocyte-macrophage lineage; this suggests that understanding monocyte-macrophage kinetics is fundamental to the study of atherosclerotic plaque biology. Atherosclerotic plaque formation is initiated by the intramural retention of atherogenic lipoproteins [51]. The accumulation of these modified lipoproteins triggers the activation of tissue-resident macrophages and the recruitment of monocyte-derived macrophages into the subendothelial space, where they engulf normal and modified lipoproteins. Although initially beneficial, these macrophages become engorged with lipids, transforming into foam cells [46]. Using the thymidine analogue 5-bromo-2′-deoxyuridine labeling and parabiosis, the role of macrophages on atherosclerosis has been found to start with circulating monocytes infiltrating into the arterial walls and differentiating to macrophages, which proliferate and locally augment their numbers in plaques [52]. Cell trafficking methods, including the labeling of circulating monocytes with fluorescent beads [53], have demonstrated that macrophage retention can be reversed through a highly regulated process, the balance of retention (e.g., neuroguidance molecule netrin 1) and emigration signals (e.g., chemokine receptor pairs) [54,55,56,57]. Therefore, the plaque macrophage burden represents the balance of blood monocyte recruitment, proliferation retention, and the emigration and death of differentiated tissue macrophages. More importantly, macrophages are also the keys for reverse cholesterol transport (RCT) pathway in atherosclerosis, by which the cholesterol moves out of the cells in atherosclerotic plaques, enters the circulation and is excreted in the feces. To prevent cholesterol toxicity, macrophages or foam cells initiate free cholesterol efflux through ATP Binding Cassette Subfamily A Member 1 (ABCA1), ATP Binding Cassette Subfamily G Member 1 (ABCG1), and other cholesterol trafficking pathway following LXR activation [58]. This is the critical and initial step of RCT in atherosclerotic plaques, and provides novel therapeutic strategies for reducing proliferation and intracellular cholesterol loading of macrophages, such as the nanoparticle delivery of simvastatin or high-density lipoprotein (HDL) [59]. However, selectively targeting macrophages for reducing atherosclerosis is challenging by the fact that the dynamic processes in regulating atherosclerotic burden vary by disease stage.

The immune system also regulates inflammation-induced atherogenic damages through the activation of pro- and anti-inflammatory mediators. Interleukin (IL)-1β, IL-6, and tumor necrosis factor-α (TNF-α) are key pro-inflammatory mediators that initiate an acute response, while IL-10 and annexin A1 are cytokines actively mediating the resolution [60]. The cytokine activation directs the polarization of either M1 or M2 macrophages [61,62]. Specifically, in the presence of high levels of pro-inflammatory cytokines and modified lipoproteins, recruited macrophages polarize toward a pro-inflammatory phenotype, the M1 type macrophages [63]. This enhances their phagocytic capacity. During resolution, the macrophages polarize toward M2 type, a pro-resolving phenotype, contribute critically to efferocytosis and participate in tissue repair and extracellular matrix production. Atherosclerosis is characterized by sustained inflammation and dysregulated resolution response. The advancement of atherosclerosis is promoted by macrophages through several mechanisms. For example, the activation of vascular smooth muscle cells by macrophages leads to the increased synthesis of pro-inflammatory cytokines and extracellular matrices, which in turn promotes further lipoprotein retention and failed efferocytosis [64,65]. The local imbalance between pro-resolving mediators and pro-inflammatory factors also feedback macrophage transformation. Efferocytosis activates several anti-inflammatory and pro-resolving pathways. Macrophage-mediated efferocytosis upregulates the production of anti-inflammatory cytokines such as IL-10 and transforming growth factor-β (TGF-β) [66]. The uptake of apoptotic bodies increases the synthesis of specialized pro-resolving mediators (SPMs) and concomitantly reduces the generation of leukotrienes [67,68]. Mer proto-oncogene tyrosine kinase (MerTK), an efferocytosis receptor, regulates SPM synthesis during the inflammatory status and switches macrophage phenotypes toward either a pro-resolving or a pro-inflammatory pathway [69]. Studies using MerTK knockout mice and mice with MerTK cleavage-resistant macrophages demonstrated MerTK increase the biosynthesis of SPM lipoxin A4 by upregulating the nuclear translocation of 5-lipoxgenase, and improved inflammation resolution and plaque regression were observed. As the atherosclerotic lesion progressed, the MerTK levels on macrophage surfaces declined because of their cleavage by ADAM metallopeptidase domain 17. Moreover, in *Ldlr*^−/−^ mice with MerTK cleavage-resistant macrophages, the atherosclerotic lesions exhibited higher levels of macrophage MerTK, lower levels of soluble Mer (the cleavage product), improved efferocytosis, smaller necrotic cores, thicker fibrous caps, and higher ratios of pro-resolving to pro-inflammatory lipid mediators [70].

These studies suggest the imbalance of cytokines, macrophages, and pro-resolving mediators for efferocytosis and inflammatory resolution leads to serious athero-progression (Figure 2). Therefore, therapies restoring the balance of inflammatory cytokines could have potential to reduce the atherosclerosis. The Canakinumab Anti-Inflammatory Thrombosis Outcomes Study (CANTOS) reported that treatment with canakinumab, a monoclonal antibody against IL-1β, led to a reduction in nonfatal myocardial infarction, nonfatal stroke, and cardiovascular death in high-risk patients, which was independent with the effect of lipid-lowering agents [71]. This was the first clinical trial to show that explicitly targeting inflammation can reduce clinical cardiovascular events. Notably, an increase in fatal infection was observed which is likely due to the immunosuppressive effects of canakinumab. More recently, the Colchicine Cardiovascular Outcomes Trial (COLCOT) and the Low-Dose Colchicine (LoDoCo) 2 trial indicated that colchicine, which suppresses inflammation by preventing cytoskeletal microtubule formation and inhibiting the nucleotide-binding oligomerization domain, leucine-rich repeat, and pyrin domain containing 3 (NLRP3) inflammasome, significantly lowered the risk of ischemic cardiovascular events in patients with recent myocardial infarction and chronic coronary disease, respectively [72,73]. Same as CANTOS trial, the incidence of infection, particularly pneumonia, occurred more frequently in the colchicine-treated group. On the other hand, the Cardiovascular Inflammation Reduction Trial indicated that low-dose methotrexate, an immune suppressant, did not decrease cardiovascular events in patients with stable atherosclerosis when it did not reduce the levels of IL-1β, IL-6, and C-reactive protein [74]. The studies highlight the mechanistic diversity of inflammatory pathways in atherosclerosis, as well as the challenge involved in the exploration of their underlying mechanisms and the selection of ideal patients. These findings and their implications serve as a reference for future studies aiming at novel inflammatory pathways to prevent atherosclerosis-related cardiovascular events.

Atherosclerosis is characterized by the retention of oxidized low-density lipoprotein cholesterol (ox-LDL-C) in the arterial wall. These modified lipoproteins activate macrophages, majorly differentiated and recruited from circulating monocytes, which engulf the deposited ox-LDL to become ‘foam cells’. These foam cells would undergo two pathways, either retention or efferocytosis. Through Mer proto-oncogene tyrosine kinase (MerTK), an efferocytosis receptor, synthesis of specialized pro-resolving mediators (SPM) in macrophages is highly regulated. Once the pro-resolving signaling and cytokines are dominant, efferocytosis-mediated resolution of atherosclerosis occurs. Moreover, macrophages or foam cells could reverse atherogenesis by initiating free cholesterol efflux to circulating high-density lipoprotein (HDL) through ATP Binding Cassette Subfamily A Member 1 (ABCA1), ATP Binding Cassette Subfamily G Member 1 (ABCG1)-mediated reverse cholesterol transport (RCT). On the other hand, if proinflammatory signaling and cytokines predominate, the failure of macrophage efferocytosis and egression leads to progression of atherosclerotic plaque formation.

## 4. Gut Barrier Regulates both Local and Systemic Immune Responses

The interface partitioning the gastrointestinal microbiome from the underlying host tissue is a pivotal signaling hub between these two compartments, and breach of this interface has been linked to detrimental metabolic consequences [75]. Intestinal homeostasis is achieved when intestinal epithelium and subepithelial immune cells act in concert with the gut microbiota [76]. The intestinal barrier serves as a gatekeeper for maintaining harmonic host-microbe communications by building a physical barrier composing of inner and outer mucus layers as well as chemical and cellular barriers, such as antimicrobial peptides (AMPs), immunoglobulin A (IgA), and the subepithelial innate lymphoid cells (ILCs). In the colon, the mucus layer is the vital component that segregates the microbiota from the epithelium. The thickness and composition of the mucus layers, which are secreted by goblet cells, are critical for the prevention of pathogen- and commensal-induced inflammation and for the maintenance of a balanced commensal population [77]. In the small intestine, where the mucus layer is discontinuously secreted and lacks distinct inner and outer layers, the homeostasis is primarily maintained by the AMPs, membrane-bound pattern-recognition receptors (PRRs), secretory IgA, CD103^+^ dendritic cells, regulatory T cells, and cytokines such as IL-33, IL-10, and TGF-β [78]. Besides, the AMPs produced by Paneth cells (a specialized epithelial cell in small intestine) such as defensins and C-type lectins, form a biochemical barrier to avoid direct contact between the host cells and the gut microbiota [79,80]. Loss of host–bacteria segregation in RegIIIγ^−/−^ mice (RegIIIγ is a secreted antibacterial lectin) increased bacterial colonization of the intestinal epithelial surface and enhanced the activation of intestinal adaptive immune responses by the microbiota [79]. TLRs and nuclear oligomerization domain (NOD)–like receptors (NLRs) are crucial pattern recognition receptors (PRRs) for identifying microbe-associated molecular patterns (MAMPs) such as LPS and peptidoglycan to regulate gut immune responses and may shape the gut microbiota. For example, studies have indicated that TLR5 deficiency leads to an increased translocation of commensals to the liver and spleen, an increased susceptibility to colitis and metabolic disorders [81,82], and long-term alterations in microbial composition. These findings suggest that TLR deficiency results in a barrier defect in commensal containment which highlights the importance of TLRs for both maintaining the gut barrier and shaping microbiota composition.

ILCs, innate immune cells without lineage markers, rapidly respond to epithelium-derived cytokine signals and are critical in maintaining intestinal homeostasis [83,84]. Found in both lymphoid and nonlymphoid tissues, ILCs, which are primarily tissue-resident cells, are particularly abundant in submucosa of the intestine and lungs but are extremely rare in peripheral blood [85,86]. ILCs are divided into three major groups—namely ILC1s, ILC2s, and ILC3s—based on their phenotype, developmental pathways, and the signature cytokines they produce [84]. ILC3s, the most abundant subset of ILCs in both the fetal and adult human intestine, produce IL-22 and IL-17 in response to the intestinal pathogen through the aryl hydrocarbon receptor [87,88]. IL-22 helps maintain the integrity of the intestinal barrier by stimulating AMPs secretion from epithelial cells and mucus production from goblet cells [89,90,91]. Mice deficient in retinoic acid receptor-related orphan receptor-γt, leading to defective ILC3, produced insufficient IgA for maintaining homeostasis of host–microbe interactions [92], and exhibited high titers of immunoglobulin G antibodies in the blood for the gut bacteria, implying a breach of the intestinal barrier may have occurred to cause systemic immune response to the microbiota [93]. A study using *Rag1*^−/−^ mice lacking mature B and T lymphocytes demonstrated that ILC3 depletion resulted in peripheral dissemination of commensal bacteria and systemic inflammation through an IL-22 blockade [94]. These findings underscore the crucial contribution of ILCs to the maintenance of intestinal homeostasis.

Dietary factors, such as high-fat and high-sugar diets can affect intestinal permeability [95,96,97]. TLRs on immune cell surfaces are the main receptors that recognize LPS, and the activation of TLR signaling pathways, particularly the TLR4/MyD88-dependent signaling pathways, induces a pro-inflammatory state [23,98]. In one landmark study, high-fat diet (HFD) modified the gut microbiome of mice and triggered an influx of bacteria-derived LPS into the systemic circulation. LPS infusion induced similar metabolic manners to that were caused by HFD, including hyperglycemia, hyperinsulinemia, and weight gain (increased explicitly fat percentages in body composition) [99]. The finding HFD disrupts gut permeability has also been validated by several other reports [100]. Interestingly, intraperitoneal administration of the LPS-TLR4 pathway inhibitor TAK-242 attenuated the effects of intestinal barrier disruption [101]. Accordingly, a homeostatic and defensive gut barrier is formed by crosstalk between the mucosal immunity, the microbiota, and the intestinal epithelia that is rigorously regulated through complex mechanisms.

## 5. Immune System Shapes Gut Microbiome in Regulating Atherogenesis

The immune system, a crucial mediator of the gut microbiota, is responsible for maintaining host-microbe homeostasis in the intestines; it must remain vigilant against invasive microbes and concomitantly limit overt inflammatory responses against commensals. Regulation of local defensive responses and modulation of systemic effects relies on the communications between the immune system and gut microbiota. The atherogenesis-associated crosstalk between the immune and nonhematopoietic cells in the arterial wall is primarily mediated by various cytokines and inflammasomes [102]. These inflammatory mediators could also regulate atherogenesis by shaping the gut microbiota.

### 5.1. IL-22 and IL-23 Signaling

IL-22, mainly produced by ILC3s in response to the microbiota, is potentiated by IL-23 and IL-1β, which are produced by inflammatory monocytes, macrophages, and dendritic cells. IL-23, a member of the IL-12 family, consists of two subunits, namely IL-12p40 and IL-23p19, and drives the mechanisms underlying several chronic inflammatory diseases [103,104]. Studies in atherosclerotic mice may present potentially protective effects for atherosclerosis through the IL-23-IL-22 axis. A study showed both global and specific deletion of the IL-23 receptor (IL-23R) in CD4^+^ T cells reduced IL-17 production but did not improve atherosclerosis in *Ldlr^−/−^* mice given a 10-week HFD [105]. In contrast, deletion of IL-23 and IL-22 in hematopoietic cells is shown to promote atherosclerosis progression without affecting IL-17 expression in the aortic plaque of *Ldlr^−/−^* mice fed with a 16-week HFD. Moreover, IL-23R deficiency in hematopoietic cells resulted in significantly increased atherosclerosis progression. Notably, the atherosclerotic phenotype was aggravated by FMT from IL-23–deficient mice when compared with wild-type (WT) mice, suggesting that inactivation of the IL-23-IL22 pathway led to atherogenesis through inducing gut dysbiosis [106]. In the same study, IL-23 and IL-22 were mechanistically demonstrated to modify the microbiota by regulating antimicrobial peptides (e.g., *Reg3b* and *Reg3g*) to restrict semi-invasive bacteria with pro-atherogenic properties. These beneficial effects may also be achieved by reducing LPS production, inhibiting TMAO biosynthesis, decreasing the expression of pro-atherogenic osteopontin [107], and inactivating Ly6C^hi^ monocytes and aortic macrophages [106]. Importantly, recent clinical studies using anti-cytokine antibodies to treat auto-immune diseases have supported the above findings. The briakinumab and ustekinumab, anti-p40 antibodies that target both IL-12 and IL-23, were used to treat patients with severe psoriasis, but significantly increased the incidence of MACE [108,109,110]. In another study, risankizumab, a selective anti-IL23p19 antibody, however, did not increase adverse cardiovascular events in patients with moderate-to-severe chronic plaque psoriasis [111]. The discrepancies in these findings can be attributed to the complexity and inter-individual variations in the interactions between the IL22-IL23 pathways and the gut microbiota. Therefore, further investigations are required to delineate the effects of the IL-22-IL-23 pathways on microbial composition and function, atheroprotection, and gut barrier maintenance.

### 5.2. Inflammasomes

Inflammasomes are cytoplasmic multiprotein signaling platforms that control the inflammatory response and coordinate antimicrobial host defense. They activate inflammatory caspases to produce cytokines and induce pyroptotic cell death [112]. Inflammasomes in the myeloid and intestinal epithelial compartments repair tissue and neutralize infection and injury partially through the maintenance of a healthy microbiota. Inflammasome activation also drives atherosclerosis development and progression through inflammatory response regulation [113].

Inflammasomes can be classified into canonical and noncanonical inflammasomes. Canonical inflammasomes serve as platforms for caspase-1 activation, whereas noncanonical inflammasomes activate caspase-11 and caspases 4 or 5 in mice and humans, respectively [114,115,116]. The noncanonical function of inflammasomes is defined as the caspase activation independent of caspase 1. Canonical inflammasome activation involves cytosolic PRRs that sense pathogen-associated molecular patterns (PAMPs) or damage-associated molecular patterns (DAMPs). Inflammasome assembly, which is hierarchical, requires mainly a sensor protein, an adaptor protein, and an effector protein [117]. Once inflammasome formation is initiated, the assembly strongly amplifies the initial activation signal [118]. Apoptosis-associated speck-like protein containing a caspase recruitment domain (ASC) and effector protein caspase 1 is critical to inflammasome assembly, the response to pyroptosis, and the production of IL-1β and IL-18.

Although NLRP inflammasomes are highly homologous, various inflammasomes may have different roles in responding to a disease entity or stimulus. Both NLRP3 and NLRP6 are crucial regulators of intestinal homeostasis, but their responses to microbiota vary [119,120]. NLRP6 is highly abundant in the intestinal epithelium, particularly in enterocytes and secretory goblet cells [120,121,122]. It is essential for mucosal self-renewal and proliferation and contributes critically to the maintenance of intestinal homeostasis and a healthy intestinal microbiota. A study reported that the gut microbiota of ex-GF *Nlrp6^−/−^* mice enclosed in a specific pathogen-free vivarium became profoundly different from that of their ex-GF WT counterparts as early as 3 weeks from the start of spontaneous conventionalization [123]. Microbiota-modulated metabolites—including taurine, histamine, and spermine—drove NLRP6 inflammasome activation to increase IL-18–induced AMP secretion. Conversely, imbalance of AMP program caused by NLRP6 inflammasome deficiency resulted in dysbiosis, which modulated and hijacked inflammasome signaling in cohoused WT mice through fecal transfer. In the dextran sodium sulfate (DSS)-induced colitis model, deficiency of ASC, caspase 1, or NLRP6 in mouse colonic epithelial cells reduced serum levels of IL-18 but not IL-1. It also led to alterations in the fecal microbiota, which was characterized by increased numbers of Bacteroidetes (Prevotellaceae) and TM7, resulting in the development of more severe colitis, which was transferable to cohoused mice [121]. However, the role of the NLRP3 inflammasome in intestinal inflammation remains unclear. Other studies on the NLRP3 inflammasome that used the DSS-induced colitis model have reported mixed findings (i.e., exacerbation and alleviation of colitis) [124,125,126]. Studies that used a mouse model of oxazolone-induced colitis or a challenge with the attaching/effacing intestinal pathogen *Citrobacter rodentium* demonstrated exacerbated colitis and increased bacterial colonization and dispersion in *Nlrp3^−/−^* mice [127,128]. Some of this evidence suggests that after colitis occurs, the production of IL-1β, IL-18, and anti-inflammatory cytokines IL-10 and TGF-β decreases in *Nlrp3^−/−^* mice, which may in turn hinder repair mechanisms and increase intestinal epithelial permeability.

Because the NLRP3 inflammasome is connected to metabolic disturbances and inflammation, it is the focus of most studies on inflammasomes related to atherogenesis. NLRP3, ASC, caspase 1, IL-1β, and IL-18 are more expressed in human carotid atherosclerotic plaque tissue than in nonatherosclerotic mesenteric or iliac arteries [129,130]. The NLRP3 inflammasome is activated by cholesterol accumulation, especially that of crystalline cholesterol, which increases the production of IL-1β and IL-18 [131,132]. A study indicated that deficiency in NLRP3, ASC, IL1α, or IL1β in hematopoietic cells significantly reduced atherosclerosis in *Ldlr**^−/−^* mice [132]. Moreover, significantly reduced atherosclerotic plaque was observed in *Ldlr**^−/−^* mice transplanted with the bone marrow of *Caspase-1/11**^−/−^* mice compared with that in *Ldlr**^−/−^* mice transplanted with control bone marrow [133]. However, *Ldlr^−/−^* mice receiving FMT from *Caspase-1/11^−/−^* mice donor exhibited progressed atherosclerosis, suggesting the gut microbiota could be further shaped by the intestinal inflammasome to regulate atherosclerosis. Introduction of the microbiota of *Caspase 1^−/−^* and *Ldlr^−/−^* mice into *Ldlr^−/−^* mice under a 13-week HFD resulted in decreased abundance of SCFA-producing bacteria, thereby increased atherosclerotic plaque formation [134]. In a mouse model of nonalcoholic fatty liver disease induced by a methionine- and choline-deficient diet, NLRP6 and NLRP3 inflammasomes and IL-18 changed the configuration of the gut microbiota to negatively regulate hepatic steatosis progression and inflammation by inhibiting the influx of TLR4 and TLR9 agonists (e.g., LPS and flagellin) into the portal circulation. Furthermore, intestinal activation of NLRP6 induced mucus layer formation by maintaining mucus granule exocytosis [120]. To protect the colonic crypt from bacterial invasion, sentinel goblet cells localized to its entrance responded to TLR agonists to activate NLRP6 inflammasome-mediated Mucin 2 secretion to segregate the bacteria [135]. In summary, the inflammasome serve as an important regulator for microbiota homeostasis and vascular health.

## 6. Gut Microbiota Regulates Immune Response in Atherogenesis

The gut microbiota plays a vital role in the immune system by regulating the development of gut-associated lymphoid tissues (GALT), including Peyer’s patches, isolated lymphoid follicles, and mesenteric lymph nodes. The gut microbiota is crucial in the maturation of innate and adaptive immune system. It has been shown that the GF mice display immature lymphoid tissues, impaired T and B cell functions, less CD4^+^ T cell counts and reduced immunoglobulin production, which make the GF mice less effective in controlling the infection [136]. Intestinal microorganisms may reside in the gut lumen stably under mucosal tolerance, in which the alteration of gut microbial community may become an inducer for some chronic inflammatory diseases such as atherosclerosis [137,138,139]. It has also been shown that antibiotics treatment disrupted the gut barrier, altered the gut microbial composition, reduced concentration of SCFA and activated NLRP3 inflammasome [140]. Taken together, these findings suggest that human colonic commensal bacteria may have an atheroprotective effect by preventing gut barrier dysfunction and the relevant inflammation.

In patients with CAD, the ratio of Firmicutes to Bacteroidetes, which constitute more than 90% of the bacterial taxa in the human gut, was shown to be higher than in that in the healthy control, and the population of Bacteroidetes, particularly *Bacteroides vulgatus* and *Bacteroides dorei*, was lower [16,141]. The abundance of *B. vulgatus* and *B. dorei* was negatively correlated with fecal LPS levels in patients with CAD. Oral gavage of *ApoE^−/−^* mice with live *B. vulgatus* and *B. dorei* substantially reduced both fecal and plasma concentrations of LPS and attenuated atherosclerotic lesion formation [141]. Moreover, gavage of HFD-treated *ApoE^−/−^* mice with live *Akkermansia muciniphila* ameliorated intestinal permeability, reduced fecal and circulating LPS, and improved aortic atherosclerosis which was independent of lipid metabolism [142]. Mechanistically, it showed that administration of *A. muciniphila* prevented mucus layer thinning in mice treated with HFD and increased the expression of occludin and ZO-1, two tight junction proteins, in the ilium.

Signals derived from gut microbiota are believed to modulate the differentiation and functional activity of macrophages in perivascular tissue and atherosclerotic plaque. Macrophages could be regulated by gut microbiota-derived components and metabolites, such as TMAO and SCFAs [143,144]. Moreover, the colonic macrophage compartment was found to be dependent on constant influx of classic Ly6C^hi^ from the circulating monocytes, and the process is primarily driven by the gut microbiota [145,146]. Lower numbers of both monocyte-derived and tissue-resident macrophages were found in GF mice. Single-cell RNA sequencing revealed that commensal microbiota affected the development, gene expression, and diversification of colonic macrophages, especially those of two populations (i.e., CD11c^+^ CD206^int^CD121b^+^ and CD11c^-^CD206^hi^CD169^+^) with unique spatial and functional roles localized within the lamina propria [147]. In response to external threats, these CD11c^+^CD206^int^CD121b^+^ macrophages expressed genes with immune effector functions. The CD11c^-^CD206^hi^CD169^+^ macrophages focused primarily on cell recruitment, cell scavenging, and tissue regeneration, which are particularly important for atherosclerotic plaque formation. The gut microbiota may affect intestinal lacteal lipid absorption through the regulation of villus macrophages. A study demonstrated that microbiota depleted with antibiotics reduced the secretion of vascular endothelial growth factor C (VEGF-C) from intestinal villus macrophages upon a MyD88-dependent pathway. Decrease in VEGF-C resulted in regression and loss of lacteal integrity, and subsequent reduction in lipid absorption, impairment of the immune surveillance system, and dysfunction of the intestinal barrier [148,149,150]. Other studies have indicated that defective lymphatics impaired macrophage reverse cholesterol transport (RCT) from atherosclerotic plaque and increase levels of atherogenic lipoproteins [151,152]. Altogether, these findings show that the gut microbiota modulates both the abundance and function of macrophages to regulate intestinal integrity and transmit the immune responses at distal organs. In addition, the host-microbe communication affects not only resident macrophages in the intestine but also those in distal organs. For example, one study reported the depletion of the gut microbiota reduced the phagocytotic capacity of alveolar macrophages and circulating neutrophils and their cellular responsiveness to LPS, leading to increased bacterial dissemination, inflammation, organ damage, and severity of pneumococcal pneumonia when compared with controls [153]. Other studies have found that gut microbiota-derived LPS promotes macrophage accumulation in the adipose tissue and increases the proliferation of resident macrophages in adipose tissue via a CD14-dependent pathway [154,155]. In a *ApoE^−/−^* mouse model, active immunization against the outer membrane protein of *Klebsiella pneumonia*, which is present in the gut, enhanced local and systemic immune control, ameliorated HFD-associated inflammation, reduced inflammatory cell numbers, and increased the polarization of M2 macrophages in the atherosclerotic plaque [156].

These findings demonstrate the importance of the crosstalk between the gut microbiota and the immune response. Studies have indicated that inflammasome activation by PAMPs and DAMPs leads to the production of IL-1β and IL-18, which enhances T_H_17 cell differentiation and the activity of IL-22 and several chemical inflammatory mediators [157,158]. In turn, these inflammatory mediators and cytokines can shape the microbiota composition and remodel the immune responses. In summary, both local and systemic responses from the interactions between gut microbiota and the immune system help modulate chronic inflammatory diseases, such as atherosclerosis.

## 7. Microbial Metabolites and Atherosclerosis

A large body of evidence has pointed that the gut microbiota conveys messages to the host through their bioactive metabolites, such as SCFAs, Bas, and TMAO, which are commonly produced via diet-microbiota-host metabolism [149]. The metabolites produced in the gut lumen can serve as signals to react at GALT in the submucosa or transmitted to regulate immune cells residing in distal tissues [159]. The immune cells primed in GALT might produce circulating cytokines or migrate to the periphery to affect systemic immunity, including the immunopathology of atherosclerosis [160]. That is to say, at least some microbial metabolites may serve as immunomodulators to promote or ameliorate atherosclerosis. Therefore, the bioactive microbial metabolites produced in the gut may become a key to decipher inflammatory mechanisms for atherosclerosis and provide opportunities in the development of novel therapeutics (Table 1) [161].

### 7.1. Bile Acids (BAs)

Primary BAs are synthesized and conjugated with glycine or taurine in the liver. After food ingestion, BAs are expelled from the gallbladder to the intestine to facilitate lipid emulsification and the absorption of dietary lipids and fat-soluble vitamins. In the intestines, the primary BAs are transformed to secondary BAs by a series of microbe-mediated processes, including deconjugation, dehydrogenation, dihydroxylation, and epimerization [162,163]. These processes transform the BAs into a hydrophobic form that is eliminated through the feces, a major route of biliary cholesterol excretion. Microbial regulation of BA metabolism can affect cholesterol and fat metabolism as well as the development of CVD [186].

BAs also function as signaling molecules that regulate glucose and lipid metabolism by binding to the nuclear receptors FXR and TGR5 [164]. Elevated levels of numerous plasma 12α-hydroxylated BAs—including taurocholic acid, taurodeoxycholic acid, glycodeoxycholic acid, deoxycholic acid, and 3-ketodeoxycholic acid—are associated with increased insulin resistance in type 2 diabetes [165]. This is supported by association between increased plasma levels of 12α-hydroxylated BAs and insulin resistance in a clinical study [166]. A study found that acarbose, a type 2 diabetes medication, modulates the gut microbiota by increasing the abundances of *Lactobacillus* and *Bifidobacterium* and depleting *Bacteroides*, therefore changing the abundance of microbial genes involved with BA metabolism.

In turn, gut microbial composition is modulated by BAs through both direct antimicrobial properties [187,188] and indirect microbial properties—through FXR-induced antimicrobial peptides [189]. Moreover, BAs exert a feedback loop to control its synthesis from cholesterol, and different levels of BAs would favor outgrowth of different microbiome, all of which produce toxic end products [190,191]. Using *Fxr* deficient and GF mice models, it has been shown that both the gut microbiota and FXR regulated the development of HFD-induced obesity adipose inflammation [192]. The effects of the gut microbiota led to metabolic syndrome, pro-inflammatory response in white adipose tissue, and increased hepatic steatosis in these mice. These conditions were less severe in the *Fxr^−/−^* mice. Notably, FXR-mediated obesity and the metabolic phenotypes were reproduced by fecal transplantation from conventionally raised WT mice to GF mice. This is supported by the positive correlation between serum low-density lipoprotein cholesterol and a high transhepatic bile-acid reflux, which results in FXR-α activation [167]. However, it has also been showed that FXR-α activation could reduce serum triglyceride levels, such as in metabolic syndrome, diabetes mellitus, or obesity [168,169]. Nevertheless, FXR-null mice in atherogenic mice models showed conflicting findings [193,194,195], which reflect the variety of BA composition, complexity of BA metabolism, and delicate balance of BA on shaping microbiota composition. This also underscores the contribution of FXR to regulate adiposity and metabolic syndrome development through alterations in microbiota composition [192,196,197].

TGR5, another BA-responsive receptor involved in metabolic control, especially glucose metabolism, is activated mainly by secondary BAs [198]. Activation of the BA–TGR5–cyclic AMP–D2 signaling pathway increases energy expenditure in brown adipose tissue, preventing obesity and insulin resistance [199]. In one study, TGR5 signaling regulated glucose homeostasis by inducing the release of intestinal glucagon-like peptide-1 (GLP-1) release in obese mice [200]. These findings indicate the atheroprotective effects of BAs through the TGR5 receptor. BA inhibited macrophage NLRP3 inflammasome via TGR5 signaling-mediated ubiquitination to ameliorate HFD-induced glucose intolerance and insulin resistance, highlighting the coordination of microbiota and immune response to regulate metabolic diseases and further atherosclerosis [201].

### 7.2. Trimethylamine N-Oxide (TMAO)

TMAO, a well-established atherogenic microbial metabolite, is derived from dietary choline and carnitine, which are metabolized to TMA by the gut flora via microbial gene cluster such as the choline utilization (*cut*) genes. TMA is oxidized to TMAO by hepatic flavin monooxygenase 3 (FMO3) [24]. Elevated TMAO has been correlated with early atherosclerosis [171], higher prevalence of CVD [172], increased risk of MACE [26], as well as more severe heart failure status [202], and higher long-term mortality risk in individuals with heart failure [202,203]. Choline, TMAO, and betaine, which are metabolites of phosphatidylcholine, predict CVD risk in humans [24]. In a study of hyperlipidemic mice transgenically expressing human ApoE-Leiden and human cholesteryl ester transfer protein, serum TMAO level was positively correlated with atherosclerosis [204]. Other studies have reported the concomitant suppression of the gut microbiota and inhibition of TMAO-induced atherosclerosis [24,25]. Consistent with the premise that TMAO contributes to atherogenesis, aortic atherosclerotic lesions increased when *ApoE^−/−^* mice received microbiota transplantation from a strain of mice producing high levels of TMAO [205].

The atherogenic mechanisms of TMAO remain unclear. According to a study in which gut microbiota suppression significantly inhibited dietary choline-induced foam cell formation and atherosclerosis in rodents, TMAO may upregulate multiple macrophage scavenger receptors linked to atherosclerosis [24]. Moreover, it inhibited macrophage reverse cholesterol transport and downregulated HDL cholesterol expression. FMO3 expression levels were negatively correlated with levels of plasma HDL cholesterol in a murine model [24], and dietary supplementation with TMAO reduced macrophage reverse cholesterol transport, an important mechanism for plaque regression [25]. TMAO also induces vascular inflammation, both in endothelial and smooth muscle cells, through activation of the mitogen-activated protein kinase and nuclear factor (NF)–κB signaling cascade [206]. Direct exposure of platelets to TMAO resulted in enhanced sub-maximal stimulus-dependent platelet activation, attributing in modulating platelet hyperresponsiveness and thrombosis risks [173].

### 7.3. Short-Chain Fatty Acid (SCFA)

SCFAs (e.g., acetate, propionate, and butyrate) derived from dietary fiber fermentation in the colon, which occurs through multiple metabolic pathways [27,174]. SCFAs transactivate and bind to PPAR-γ [175], which increases energy metabolism and regulates lipid metabolism [207]. SCFAs also play essential roles as substrates for the central carbon, glucose, lipid, and cholesterol metabolisms [181]. They regulate glucose metabolism by triggering the secretion of incretin hormone GLP-1 through the activation of free fatty acid receptor 2, which is coupled to G protein, on L cells [182]. GLP-1 release is also involved in appetite regulation, body weight maintenance, and adiposity distribution [180].

In vascular smooth muscle cells and the juxtaglomerular apparatus, SCFAs inhibit renin release and induce vascular relaxation through G-protein-coupled receptors such as olfactory receptor 78 to reduce blood pressure (BP) [176]. This effect was reversed by antibiotic-mediated depletion of the SCFA pool. This finding was supported by a clinical trial that demonstrated the BP-lowering effects by dietary fiber [208] and another study that proved transferable hypertensive phenotypes by FMT [209]. Another study indicated that butyrate, specifically the implantation of butyrate-producing bacterium *Roseburia intestinalis*, reduced atherosclerotic lesions in both *ApoE^−/−^* mice and humanized gnotobiotic mice. Butyrate enhances gut barrier function and reduces systemic inflammation, including macrophage plaque infiltration, without affecting levels of serum cholesterol or TMAO [178]. In a human epidemiology study, dietary fiber consumption reduced the risk of CVD, a finding that supports the premise that SCFAs alleviate atherosclerosis [179]. Dietary fiber and SCFA acted on G-protein coupled receptor GPR43 of colonic epithelial cells to activate the NLRP3 inflammasome and further ameliorate DSS-induced colitis through IL-18-mediated gut homeostasis [177]. Interestingly, another inflammasome, NLRP1, aggravated DSS-induced colitis by reducing SCFA-producing *Clostridiales* [210]. SCFA promotes IL-22 production by CD4+ T cell and ILCs, both in vitro and in vivo, by upregulating AhR and HIF-1α expression to inhibit intestinal inflammation [211]. Therefore, SCFAs are the pivotal mediators between microbiota and immune response and could be a therapeutic target for immunomodulation.

### 7.4. Phenylacetylglutamine (PAGln)

PAGln, which is derived from phenylalanine by the colon microbiota [183], is a newly identified microbial metabolite correlated with MACE in patients with and without diabetes [29]. In the colon, the gut microbiota possesses pyruvate ferredoxin oxidoreductase A, which converts phenylalanine into phenylacetic acid, and then PAGln and phenylacetylglycine [183,184]. Murine models of FeCl_3_-induced carotid artery injury have indicated that PAGln provides cellular responses through the α2A-, α2B-, and β2-adrenergic receptors (ADR) [29], the principal regulators of cardiovascular function and platelet aggregation [185,212,213]; this activates platelets and induces thrombosis potential in whole-blood and isolated platelets. Administration of β-blocker (carvedilol) attenuated PAGln-triggered ADR signaling events and thrombosis in vivo [29], echoing the clinical discovery of the inhibitory effects of carvedilol on platelet aggregation [214]. The evidence indicates that PAGln is a promising target for thrombosis and atherosclerosis treatment.

## 8. Conclusions

Microbiota are essential immune modulators to promote human health. Dysbiosis results in both local inflammatory response, such as inflammatory bowel disease, and pro-inflammatory status in distal organs/tissues, such as atherosclerosis. Microbiota and their metabolites are widely studied in systemic inflammatory diseases and atherosclerosis. Until now, however, application of antibiotics, probiotics, or prebiotics in order to reshape microbiota composition has yet to be demonstrated as therapeutic options for atherosclerosis. It is probably because the complexity of interactions between microbiota and immune responses hinder this therapeutic attempt. Recent basic researches have indicated specific cytokines (such as IL-22 and IL-18), inflammasome (such as NLRP3 and NLRP6), and microbial metabolites (such as TMAO, PAGln, SCFAs) might be potential pharmacological targets (Figure 3). Recent important clinical trials, CANTOS, COLCOT, and LoDoCo2, also provide promising evidences by using immunomodulation as a novel therapeutic approach for treating atherosclerosis. Through better understanding of the dynamic and complex interplay between microbiota and immune responses (Table 2), both locally and systemically, a new layer of therapeutic strategies to treat atherosclerosis may emerge in the near future.

Gut barrier composes of commensal gut bacteria, mucus layer, innate lymphoid cells (ILC) and antimicrobial peptides (AMP). Immune cells, such as neutrophils and macrophages, cytokines, such as IL-18 and IL-22, and inflammasomes, such as NLRP3 and NLRP6, help shape the composition of gut microbiota, which in turn keeps metabolic homeostasis and inhibits inflammation. Importantly, these immune cells and cytokines orchestrate with microbiota and its metabolites to regulate atherogenesis and plaque regression. When dysbiosis occurs (right panel), structural components of dead bacteria, e.g., LPS, microbiota-derived functional metabolites, e.g., trimethylamine N-oxide (TMAO) and phenylacetylglutamine (PAGln), and inflammatory cytokines are released into local micro-environment and systemic circulation to induce several atherogenic reactions, such as inhibition of reverse cholesterol transport (RCT), hypercholesterolemia, platelet activation, and inflammatory response. Dysbiosis also impairs lacteal integrity and inhibits RCT by reducing the secretion of vascular endothelial growth factor C (VEGF-C), an important regulator of lymphatic vessel growth, from intestinal villus macrophages. (LPS: lipopolysaccharide, SCFA: short-chain fatty acid).

## Figures and Tables

**Figure 1 ijms-21-08729-f001:**
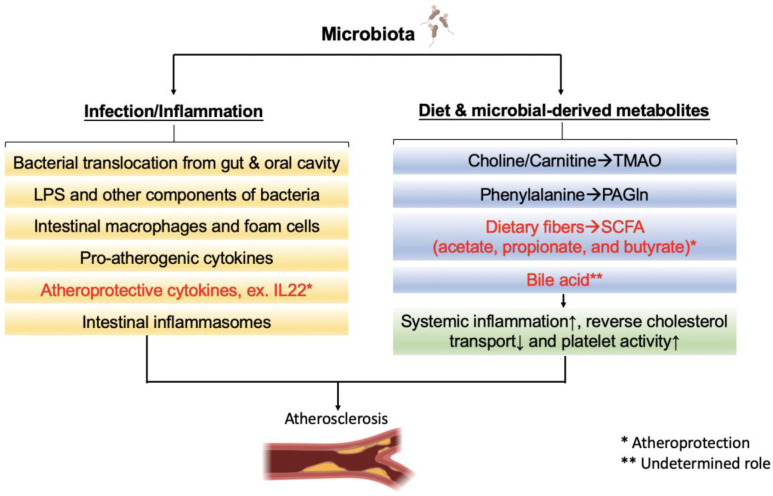
Microbiota-related pathways in atherosclerosis.

**Figure 2 ijms-21-08729-f002:**
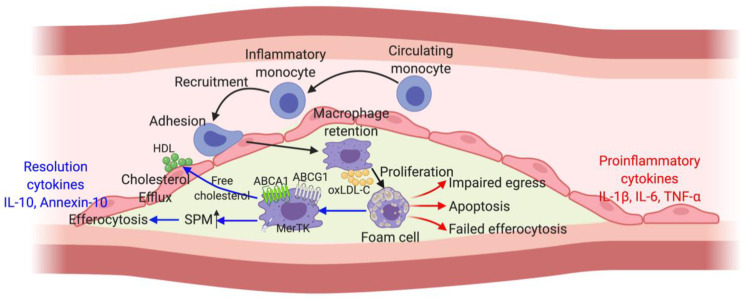
Role of macrophages in atherosclerosis.

**Figure 3 ijms-21-08729-f003:**
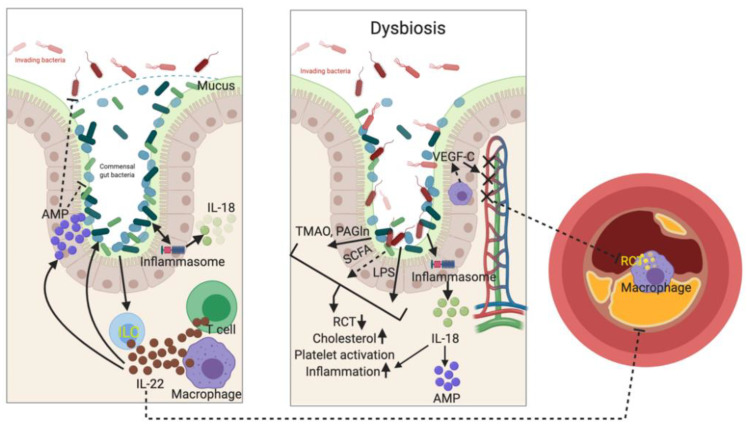
Mutual interplay of microbiota and immune system on atherogenesis.

**Table 1 ijms-21-08729-t001:** Action of microbial metabolites on atherosclerosis.

Metabolite	Precursor	Gut Microbial Metabolism	Site of Action	Atherosclerosis Effect	Physiological Action	Mechanism in Atherosclerosis
Bile acid	Primary bile acids [162,163]	Dehydroxylation, deconjugation, dehydrogenation, epimerization, [162,163]	FXR [164]	Undetermined	Increase insulin resistance Increase hepatic and serum triglyceride and cholesterol levels [165,166]	FXR-dependent [167]
			TGR5 [164]		Improve insulin resistance Increase (extra)hepatic metabolism of VLDL and fatty acid [168,169]	
TMAO	Choline L-carnitine [24]	TMA-lyase (*CutC/D* and others) [24,170]		Proatherogenic [26,171,172]	Induce vascular inflammation [24] Induce macrophage foam cell formation [24] Platelet activation [24,173]	Upregulate multiple macrophage scavenger receptors [24] Inhibit macrophage reverse cholesterol transport [25]
SCFA	Dietary fiber [27,174]	Wood-Ljungdahl pathway [27,175] Succinate pathway, propanediol pathway, acrylate pathway [27,175]	Olfr78 [176] Gpr43 [177]	Atheroprotective [178,179]	Decrease blood pressure [176] Body weight reduction [180] Decrease appetite [180] Improve liver steatosis [181]	Induce GLP-1 [180,182] Reduce migration of macrophages into plaques [178]
PAGln	Phenylalanine [183]	Pyruvate ferredoxin, oxidoreductase A (PorA) [183,184]	Adrenergic receptors [29]	Proatherogenic [29,30]	Increase platelet reactivity [29,185]	Induce thrombosis [29,185]

FXR: farnesoid X receptor, GLP-1: glucagon-like peptide-1, PAGln: phenylacetylglutamine, PPARγ: peroxisome proliferator–activated receptor-γ, RCT: reverse cholesterol transport, SCFA: short-chain fatty acid, TGR5: Takeda G-protein coupled bile acid receptor, TMAO: trimethylamine N-oxide, VEGF-C: vascular endothelial growth factor C.

**Table 2 ijms-21-08729-t002:** Summary of important studies.

Reference	Model	Aim of Study	Experiment Design	Main Finding
[24]	Human Mouse	Gut flora-dependent metabolism of dietary phosphatidylcholine on CVD pathogenesis	Metabolomics approach in human cohort Choline isotope tracer feeding in mice	Dietary supplementation with choline or TMAO promoted upregulation of macrophage scavenger receptors linked to atherosclerosis, and aggravated atherosclerosis
[25]	Human Mouse	Role of gut microbiota on TMAO production from dietary L-carnitine and relationship of TMAO and CVD risk	Metabolomics approach Human/mouse microbiota analyses Isotopic L-carnitine feeding in mice	L-carnitine supplementation significantly altered cecal microbial composition, markedly enhanced synthesis of TMA/TMAO, and increased atherosclerosis
[29]	Human Mouse Microorganism	Identifying novel pathways linked to CVD	Metabolomics approach in CVD vs. non-CVD patients In vivo FeCl_3_-induced thrombosis model	PAGln represents a new CVD-promoting gut microbiota-dependent metabolite that signals via adrenergic receptors
[55]	Mouse	Mechanism of HDL promoting regression of atherosclerosis	Aortic transplantation Lipid and Lipoprotein Analyses	HDL as a regulator of the migration and inflammation of monocyte-derived cells in murine atherosclerotic plaques
[59]	Mouse	Effect of simvastatin on macrophages and plaque regression	Nanoparticle-based delivery of simvastatin in mice with advanced atherosclerotic plaques	Pharmacologically inhibiting local macrophage proliferation can effectively treat inflammation in atherosclerosis
[71]	Human	Feasibility of reducing inflammation to decrease the risk of CVD clinically	Canakinumab 150mg every 3 months, randomized controlled and double blind trial	Antiinflammatory therapy targeting the IL-1β led to a significantly lower rate of recurrent cardiovascular events
[79]	Mouse	Role of RegIIIγ on the bacterial colonization of the mucosal surface	*RegIIIγ^−/−^* and *Myd88^−/−^* vs. wild-type littermate FISH analysis for spatial relationships between the microbiota and the host mucosal surface	RegIIIγ is a fundamental immune mechanism that promotes host-bacterial mutualism by regulating the spatial relationships between microbiota and host
[106]	Mouse	Role of IL-23 on atherosclerosis	Bone marrow deletion of IL-23 FMT	The IL-23-IL-22 signaling as a regulator of atherosclerosis that restrains expansion of pro-atherogenic microbiota
[134]	Mouse	Impact of microbiota from *Casp1^−/−^* mice on atherogenesis in *Ldlr^−/−^* mice	FMT from *Casp1^−/−^* mice to *Ldlr^−/−^* mice following antibiotics treatment	FMT of proinflammatory *Casp1^−/−^* microbiota into *Ldlr^−/−^* mice enhances systemic inflammation and accelerates atherogenesis
[142]	Mouse	Role of *Akkermansia muciniphila* in the pathogenesis of atherosclerosis	*ApoE^−/−^* mice treated with *A. muciniphila* by daily oral gavage	*A. muciniphila* attenuates atherosclerotic lesions by ameliorating metabolic endotoxemia-induced inflammation through restoration of the gut barrier
[173]	Human Mouse	Role of TMAO on platelet activity and thrombosis	In vivo FeCl_3_-induced thrombosis model Metagenomic analyses by sequencing 16S ribosomal RNA in cecal microbiota FMT Platelet activity from human samples	Gut microbes, via generation of TMAO, can directly modulate platelet hyperresponsiveness and clot formation rate in vivo
[178]	Mouse	Effect of butyrate-producing bacteria on atherosclerosis	FMT of either low or high butyrate-producing human microbiota to GF mice	Colonization with butyrate producing *R. intestinalis* decreases levels of inflammatory markers and atherosclerosis in a diet-dependent manner
[205]	Mouse	Effect of gut microbial transplantation from high to low TMAO-producing mice on atherosclerosis susceptibility	FMT from high to low TMAO-producing mice	Atherosclerosis susceptibility may be transmitted via transplantation of gut microbiota

CVD: cardiovascular disease, FISH: fluorescence in situ hybridization, FMT: fecal microbial transplantation, GF: germ-free, HDL: high-density lipoprotein, LDLR: low-density lipoprotein receptor, PAGln: phenylacetylglutamine, TMAO: trimethylamine N-oxide.

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
