# Peer review of "Mutual Interplay of Host Immune System and Gut Microbiota in the Immunopathology of Atherosclerosis"

_ijms, 2020, doi:10.3390/ijms21228729_

Round 1

Reviewer 1 Report

The aim of the manuscript is to provide a comprehensive review focused on the state-of-the art knowledge about the relationship between gut micobiome and atherosclerosis. The strong point of this review is wide coverage of the current literature and the clear organization of the text. From my point of view, the weakness of this manuscript is the accentuation of the immunological mechanisms while relatively less attention is paid to the microbiome. The manuscript is written as long text with only one table. For readers´ comfort I suggest to provide the overview of the most important studies cited in the text in the form of tables (reference - model - aim of study - exp design - main findings) or schemes.

Author Response

We thank for the reviewer’s constructive suggestion. We have added table 2 to summarize and provide the overview of the most important studies in this manuscript.

Reviewer 2 Report

In the present review, Yeh et al. discussed the current knowledge regarding to the roles of microbiota in contributing atherosclerotic pathogenesis and highlighted the mutual interplay between microbiota and immune system on atherosclerosis. This is one of the important aspect in the pathogenesis of the atherosclerosis and its adverse effects. This review has been well written and nicely discussed. However, there are some sections that need to be discussed before it can be considered for publication.

Comments:

  1. Authors should provide the schematic diagram to depict the major roles provided by the gut microbiota and immune system in the athero-protective and pro-atherogenic pathways.
  2. What is the role of reverse cholesterol transport (RCT) in the pathogenesis of the atherosclerosis, need to describe more.
  3. The involvement of commensal bacteria in atherosclerosis need to discuss.
  4. It will be more helpful if authors will provide the respective references in the Table 1.
  5. Figure 1 and the figure legends should be more descriptive.
  6. Authors may discuss a section regarding the role of B cells in atherosclerosis.

Author Response

We thank for the reviewer’s helpful comments. Our responses to these comments are as follows.

Comment 1: We have added figure 1 and figure 2 to depict the major roles of the gut microbiota and macrophages on atherosclerosis.

Comment 2: We have added more description of RCT on atherosclerosis in page 5 line 198-205 (labelled as red-colored text).

Comment 3: We have added the description of the involvement of commensal microbiota on atherosclerosis in page 2 line 83-92 (labelled as red-colored text).

Comment 4: We have added the respective references to provide more information in table 1.

Comment 5: We have added more detailed legend description for figure 1 (now figure 3) (labelled as red-colored text).

Comment 6: We thank for the reviewer’s interesting suggestion. However, so far there is limited evidence linking the role of gut microbiota and B cell biology in atherosclerosis development, even though B cells may participate in part of atherogenesis. Therefore, we did not discuss the roles of B cell in this manuscript. We believe it is a good topic and could be discussed more comprehensively when more relevant scientific data were accumulated. 

We have also corrected some mislabeling in reference citation (192) and table 1 (Gpr43 instead of Gpr41).